# Structural and functional dissection of the interplay between lipid and Notch binding by human Notch ligands

Richard J Suckling[1,†], Boguslawa Korona[2,†], Pat Whiteman[2], Chandramouli Chillakuri[2], Laurie Holt[2], Penny A Handford[2,*] & Susan M Lea[1,**]

## Abstract

Recent data have expanded our understanding of Notch signalling by identifying a C2 domain at the N-terminus of Notch ligands, which has both lipid- and receptor-binding properties. We present novel structures of human ligands Jagged2 and Delta-like4 and human Notch2, together with functional assays, which suggest that ligand-mediated coupling of membrane recognition and Notch binding is likely to be critical in establishing the optimal context for Notch signalling. Comparisons between the Jagged and Delta family show a huge diversity in the structures of the loops at the apex of the C2 domain implicated in membrane recognition and Jagged1 missense mutations, which affect these loops and are associated with extrahepatic biliary atresia, lead to a loss of membrane recognition, but do not alter Notch binding. Taken together, these data suggest that C2 domain binding to membranes is an important element in tuning ligand-dependent Notch signalling in different physiological contexts.

**Keywords** Delta-like; Jagged; lipid-binding; Notch; specificity
**Subject Categories** Signal Transduction; Structural Biology
**The EMBO Journal (2017) 36: 2204–2215**

See also: **B-Z Shilo & D Sprinzak** (August 2017)

## Introduction

The Notch signalling pathway is conserved across all metazoan species and plays key roles in many aspects of cell biology including cell-fate determination, stem cell maintenance, immune system activation and angiogenesis in humans (Guruharsha *et al*, 2012; Bray, 2016). Aberrant Notch signalling results in a number of inherited and acquired disorders, including various cancers, and it is therefore a key target for therapeutic intervention (Hansson *et al*, 2004; Groth & Fortini, 2012). Both the Notch receptors and the ligands are single-pass type I transmembrane proteins, and direct protein–protein contact between adjacent cells initiates an intracellular signalling pathway. The Notch receptor exists as a heterodimeric transmembrane protein with the N-terminal extracellular domain consisting of up to 36 tandem epidermal growth factorlike (EGF) repeats. Binding of a Notch ligand to EGF11-12 of a Notch receptor results in a series of proteolytic cleavages, with the final intramembrane cleavage by gamma secretase, causing the release of the intracellular domain of Notch (NICD) from the plasma membrane. Once released, NICD translocates to the nucleus where it binds to a transcription factor of the CBF1, Suppressor of Hairless, Lag-1 (CSL) family in complex with the coactivator MAML. This complex then relieves repression and activates Notch target genes of the *Hes* and *Hey* repressor families (Nam *et al*, 2006). Whilst *Drosophila* have one Notch receptor, mammalian species have four (Notch1-4). Notch–ligand interactions can result in activation or inhibition of Notch signalling, depending on whether ligands are presented to Notch on neighbouring cells (*trans*), or on the same cell (*cis*) (Sprinzak *et al*, 2010).

There are four canonical cell surface mammalian Notch ligands, Jagged1, Jagged2, Delta-like1 (Delta-like1) and Delta-like 4 (Delta-like4), and one non-canonical ligand, Delta-like3, which is unable to activate Notch and is found predominantly in the Golgi apparatus (Ladi *et al*, 2005; Geffers *et al*, 2007; Serth *et al*, 2015). All of the Notch ligands have a modular extracellular architecture consisting of an N-terminal C2 domain (formerly known as the MNNL domain), a Delta/Serrate/Lag-2 (DSL) domain, and either 16 (Jagged1/Jagged2), 8 (Delta-like1/Delta-like4) or 7 (Delta-like3) EGF repeats. We have previously shown that the very N-terminal domain of human Jagged1 is a C2 domain, and lipid binding of this domain is required for optimal Notch activation (Chillakuri *et al*, 2013). This domain is conserved across the Notch ligands; however, the loops within the putative lipid-binding site vary considerably between ligands, suggesting that the ligands have different lipid-binding specificity.

1  Sir William Dunn School of Pathology, University of Oxford, Oxford, UK
2  Department of Biochemistry, University of Oxford, Oxford, UK
   *Corresponding author. Tel: +44 1865 613200; E-mail: penny.handford@bioch.ox.ac.uk
   **Corresponding author. Tel: +44 1865 275500; E-mail: susan.lea@path.ox.ac.uk
   †These authors contributed equally to this work

Recently, a co-crystal structure of a Delta-like4 variant N-EGF1 in complex with Notch1 EGF11-13 (Luca *et al*, 2015) has shown that Notch ligands interact via a platform located on one side of the N-terminal C2 domain away from the lipid-binding region (site 1) and via their DSL domain (site 2), with Notch receptor domains EGF11 (site 2) and 12 (site 1) in an antiparallel fashion. This confirmed previous data showing that residues in the Jagged1 DSL domain, and in Notch1 EGF12, are critical for receptor–ligand interactions (Cordle *et al*, 2008; Whiteman *et al*, 2013). *O*-glycosylation of Notch plays an important role in regulating Notch signalling, with *O*-fucosylation on Thr-466 in EGF12 of Notch1 enhancing ligand binding (Stahl *et al*, 2008; Yao *et al*, 2011). In addition, we have also shown that Fringe-catalysed addition of GlcNAc to the *O*-fucose at Thr-466 in EGF12 increases binding to ligands (Taylor *et al*, 2014). The Delta-like4-Notch1 complex structure shows that the *O*-fucose modification directly contributes to the binding interface, in addition to specific amino acid contacts (Luca *et al*, 2015).

Here, we further highlight the variability of the C2 domain putative lipid-binding site in Notch ligands, by solving the crystal structures of N-terminal fragments of both human Delta-like4 and Jagged2. These new structures, together with a structure for Notch2 EGF11-13, have allowed a detailed comparison of both ligand/receptor and ligand/lipid-binding interactions. We further demonstrate *in vitro* that Notch receptor binding to ligand enhances interactions with lipids, suggesting that a ternary complex between Notch, ligand and lipid fine tunes generation of the Notch signal at the cell surface. A subset of *Jagged1* mutations which are associated with extrahepatic biliary atresia and affect the loops at the apex of the C2 domain reduce both Notch activation and lipid binding indicating the importance of membrane binding for tuning the Notch signal in specific physiological contexts.

# Results

### Structures of the N-terminal domains of human Notch ligands highlight conformational flexibility between EGF2-3 which may facilitate formation of an extended Notch/ligand-binding interface

We expressed and solved the structures of various N-terminal fragments of both human Delta-like4 and Jagged2 (Table 1). These fragments include the N-terminal C2 lipid-binding domain, the receptor-binding DSL domain, and two ("N-EGF2") or three ("N-EGF3") adjacent EGF domains. Here, we present the first structures of Jagged2 (N-EGF2 and N-EGF3), which has only 58% sequence identity with human Jagged1 (N-EGF3), together with the longest known structure of a Delta-like4 ligand (human Delta-like4 N-EGF3) (Fig 1A). This allows, for the first time, comparative structural analyses of all the canonical ligands. Superposition of the different Jagged2 structures from our study, with all the various Notch ligand structures (Chillakuri *et al*, 2013; Kershaw *et al*, 2015) across the DSL domain, shows that within this region the ligands form a near-linear domain organization (Fig 1B). However, the angle between adjacent domains can vary subtly, resulting in the ligand structures appearing to fall into two groups, one group including Jagged1, Delta-like1 and

some structures of Jagged2, and the other including Delta-like4, and some structures of Jagged2. The observation that Jagged2 is split across the two groups suggests that there is some flexibility between adjacent domains of the Notch ligands and that all may be able to adopt these different conformations. It is therefore likely that the conformation seen is determined by crystal packing. For example, the angle between EGF2 and EGF3 in our hDelta-like4 structure is likely due to interactions with a neighbouring molecule in the crystal lattice. The only ligand–Notch complex seen to date shows the ligand to be in the more bent conformation, and we propose that this is due to the use of a short Notch EGF11-13 construct. We have previously shown that modelling of Notch EGF4-13 in complex with ligand, based on the published structure of the receptor/ligand complex, leads to a steric clash (Weisshuhn *et al*, 2016). However, if the ligand remodels into the straighter form, shown in our new structures, this allows good packing interactions between Notch EGF4-13 and ligand, extending the binding interface along the longitudinal axis, and suggests that remodelling is a prerequisite for the ligand to form optimal contacts with Notch (Fig 1C) (See Note added in proof).

### Comparison of the N-terminal C2 domains of human Notch ligands shows differences in the lipid-binding region

All of the C2 domains of the human Notch ligands superpose with root-mean-squared deviations (RMSDs) between atomic positions of between 1.1 and 1.5 Å. All have type II topology and are most similar to members of the PKC-C2 family (Corbalan-Garcia & Gomez-Fernandez, 2014) including Munc-13 and phospholipase A2 (cPLA2) (from a protein structure-based DALI search http://ekhidna.bioce nter.helsinki.fi/dali_server/start). One distinct feature of the Notch ligand C2 domains is the presence of a long loop between strands 2 and 3, which forms the Notch EGF12 binding site 1. The stability of this loop is supported by a disulphide bond between the loop, and strand 2 (Luca *et al*, 2015). Despite strong overall structural conservation of the C2 domains, there are major differences in the loops between strands 1 and 2, between strands 3 and 4 and between strands 5 and 6. This region, at the extreme termini of the Notch ligands, is the putative lipid-binding site (Fig 2A and B), and the differences in the loops of the ligand C2 domains most likely confer different lipid-binding specificities. Even between rat and human Delta-like4, there are loop differences, suggesting that these regions are optimized for the different mammalian physiologies (Fig EV1).

The C2 loops are ordered upon calcium binding in both Jagged1 (Chillakuri *et al*, 2013) and Jagged2 and contain the amino acid ligands for calcium binding although the number of calcium ions bound differs. Interestingly, the three calcium ions bound in the Jagged2 C2 domain are at equivalent positions to three of the five calcium ions bound in the Perforin C2 domain (Yagi *et al*, 2015). The loops are fully ordered in Delta-like4, and mostly ordered in Delta-like1 (Kershaw *et al*, 2015) despite the absence of calcium-binding sites in the Delta family (Fig 2A and B). Consistent with this, two of the key aspartate side-chain calcium ligands conserved in Jagged1 and 2 are replaced with arginine and histidine residues (Arg-55 and His-123 in Delta-like4) (Fig 2C). All of the Notch ligands, irrespective of whether they have calcium-binding sites or not, contain few hydrophobic residues in calcium-binding region 1

**Table 1. Data collection and refinement statistics.**

| | Human Jagged2 | | | | |
| --- | --- | --- | --- | --- | --- |
| | Jagged2 N-EGF2 | Jagged2 N-EGF2 | Apo-Jagged2 N-EGF3 | DLL4 N-EGF3 | Notch2 EGF11-13 |
| **Data collection** | | | | | |
| Beamline | Diamond I02 | Diamond I03 | Diamond I04-1 | Diamond I04-1 | Diamond I04 |
| Space group | $P2_12_12_1$ | P1 | $P2_12_12_1$ | C2 | $P2_12_12_1$ |
| Wavelength (Å) | 1.3 | 0.93 | 0.92 | 0.92 | 0.98 |
| Cell dimensions (Å) | | | | | |
| $a, b, c$ (Å) | 48.3, 83.9, 99.4 | 62.6, 92.9, 97.0 | 46.8, 77.2, 96.4 | 127.9, 49.8, 70.0 | 20.2, 49.8, 125.5 |
| $\alpha, \beta, \gamma$ (°) | 90.0, 90.0, 90.0 | 71.7, 83.2, 82.7 | 90.0, 90.0, 90.0 | 90.0, 109.2, 90.0 | 90.0, 90.0, 90.0 |
| Resolution range (Å)[a] | 83.9–2.70 (2.83–2.70) | 38.0–2.80 (2.89–2.80) | 77.2–2.80 (2.97–2.80) | 66.2–2.17 (2.24–2.17) | 41.8–1.86 (1.91–1.86) |
| $R_{merge}$[a, b] | 0.107 (0.559) | 0.103 (0.685) | 0.267 (1.116) | 0.054 (0.508) | 0.077 (1.067) |
| $R_{meas}$[a, c] | 0.115 (0.598) | 0.134 (0.891) | 0.299 (1.246) | 0.062 (0.594) | 0.096 (1.380) |
| $CC_{1/2}$[a, d] | 0.996 (0.892) | 0.990 (0.580) | 0.977 (0.648) | 0.998 (0.852) | 0.998 (0.524) |
| Mean $I/\sigma I$[a] | 12.9 (3.7) | 7.3 (1.3) | 6.2 (1.5) | 14.6 (2.3) | 11.2 (1.3) |
| Completeness (%)[a] | 99.6 (100.0) | 97.6 (97.7) | 99.4 (99.7) | 98.0 (91.6) | 99.5 (99.3) |
| Multiplicity[a] | 7.7 (8.1) | 2.2 (2.3) | 4.8 (5.0) | 4.5 (3.7) | 5.2 (5.1) |
| Wilson <B> (Å$^2$) | 48.9 | 50.0 | 37.1 | 42.5 | 31.1 |
| **Refinement** | | | | | |
| Resolution range (Å) | 64.1–2.70 | 38.0–2.80 | 60.3–2.80 | 66.2–2.17 | 39.0–1.86 |
| No. of reflections | 11,513 | 49,334 | 8,975 | 21,802 | 11,247 |
| $R_{work}/R_{free}$ | 0.2299/0.2635 | 0.2172/0.2751 | 0.2550/0.3121 | 0.1823/0.2293 | 0.2278/0.2636 |
| **Number of atoms** | | | | | |
| Protein | 2,030 | 12,871 | 2,307 | 2,321 | 918 |
| Ligand/ion | 3 | 19 | 0 | 0 | 3 |
| Water | 43 | 232 | 35 | 196 | 121 |
| B factors (Å$^2$) | 55.6 | 51.3 | 40.1 | 49.5 | 38.8 |
| **Rmsd from ideal values** | | | | | |
| Bond lengths (Å) | 0.006 | 0.013 | 0.004 | 0.005 | 0.021 |
| Bond angles (°) | 0.686 | 0.755 | 0.662 | 0.812 | 0.550 |
| **Ramachandran plot** | | | | | |
| Favoured region (%) | 92.5 | 91.7 | 87.3 | 95.3 | 98.3 |
| Allowed (%) | 100.0 | 99.7 | 97.9 | 100.0 | 100.0 |
| Outliers (%) | 0 | 0.3 | 2.1 | 0 | 0 |
| Rotamer outliers (%) | 0.9 | 0.3 | 1.2 | 0 | 0 |
| C-beta outliers | 0 | 1 | 0 | 0 | 0 |
| PDB ID code | 5MW5 | 5MWF | 5MW7[#] | 5MVX | 5MWB |

[a]Values in parentheses are for the highest resolution shell.

[b]$R_{merge} = \frac{\sum_{hkl}\sum_j |I_{hkl,j} - <I_{hkl}>|}{\sum_{hkl}\sum_j I_{hkl,j}}$ where $<I_h>$ is the mean intensity of unique reflection $h$, summed over all reflections for each observed intensity $I_{hl}$.

[c]$R_{meas} = \frac{\sum_{hkl}\sqrt{\frac{n}{n-1}}\sum_{j=1}^{n} |I_{hkl,j} - <I_{hkl}>|}{\sum_{hkl}\sum_j I_{hkl,j}}$ where $n$ is the number of observations for unique reflection $h$ with mean intensity $<I_h>$, summed over all reflections for each observed intensity $I_{hl}$.

[d]$CC_{1/2}$ is the correlation coefficient on <I> between random halves of the dataset. $\Delta_{anom}$, anomalous difference $I^+ - I^-$.

(CBR1) or CBR3, and those that are present are not at the tip of the loops (tip being defined as closest to the membrane) (Fig 2). This suggests that the Notch ligand C2 domains are not deeply buried within the membrane upon binding and are therefore distinguishable from intracellular C2 domain proteins, and Perforin (Yagi *et al*, 2015).

[#]Correction added on 1 August 2017 after first online publication: PDB code "5MV7" was corrected to "5MW7".

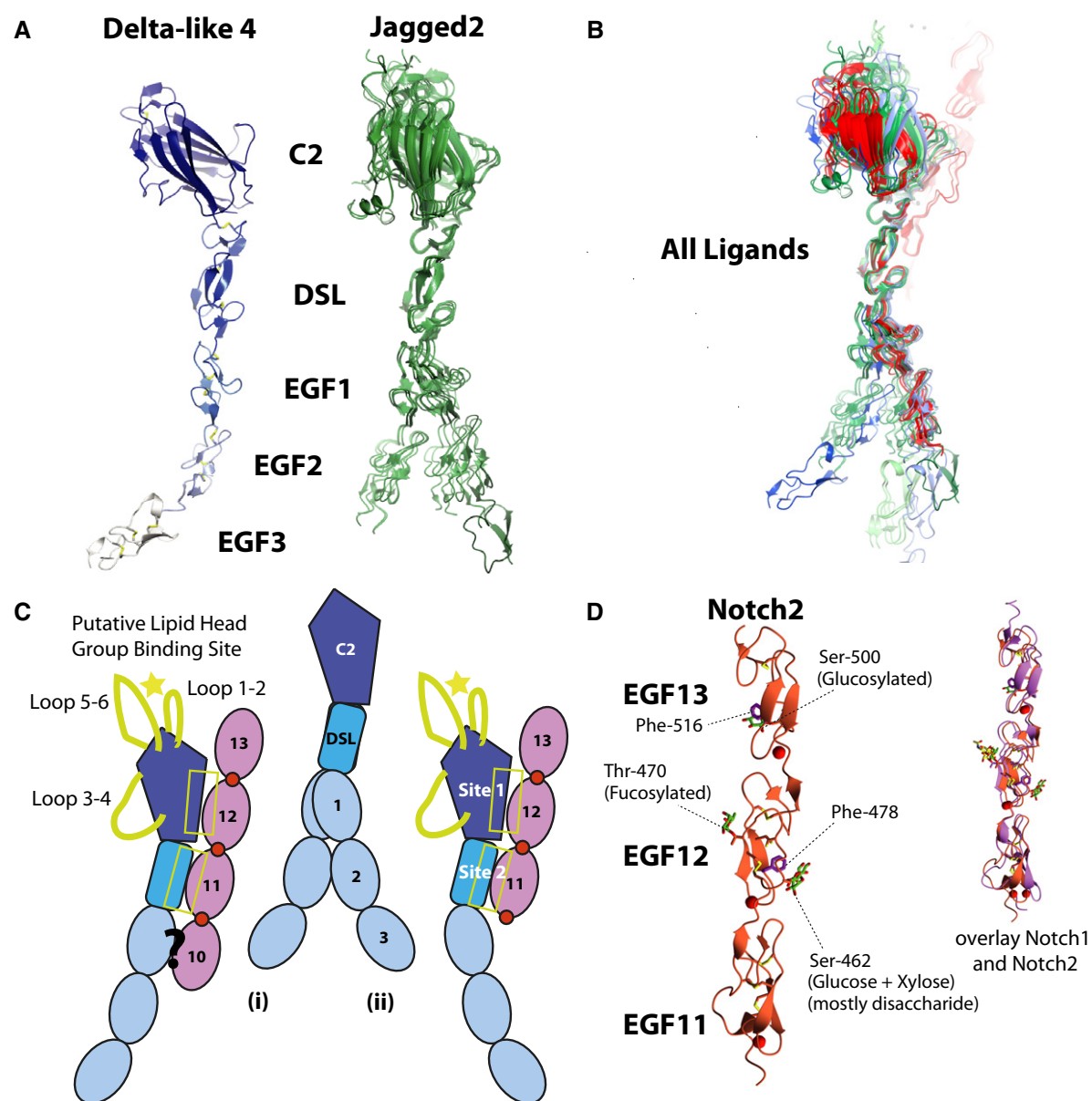

**Figure 1.   New structures of human Notch ligands and receptors.**

A     Structures of human Delta-like4 (N–EGF3) and human Jagged2 (N–EGF2 and N–EGF3). Two crystal forms of Jagged2 (N–EGF2) (green) and one crystal form of Jagged2 (N–EGF3) (dark green) were solved, including a total of seven crystallographically independent copies of N–EGF2, and 1 copy of N–EGF3 (N.b. not all of EGF3 is visible in the electron density). None of the EGF domains are of the calcium binding type. All of these copies have been superposed on each other by alignment of the DSL domain, showing some flexibility in the hinge region between the C2 and DSL domains, and further flexibility in the hinge regions between the EGF domains.

B, C     Superposition of all of the known human Notch ligand structures (Jagged1, PDB ID = 4CC1 (light green) (Chillakuri *et al*, 2013), Delta-like1, PDB ID = 4XBM (light blue) (Kershaw *et al*, 2015), Delta-like4 (blue), Jagged2 (green)), and all of the Delta-like4 variant-Notch1 complex structures (red) across the DSL domain shows that these structures appear to fall into two groups (C). One group has an overall globally bent arrangement (C(ii)), which includes all of the Delta-like4 molecules bound to Notch1 EGF11-13, and a second group with a straighter arrangement (C(i)). Binding of ligands to a Notch receptor in a native context likely requires the ligands to be in the straighter arrangement, as the bent arrangement is incompatible with binding to EGF10 (Weisshuhn *et al*, 2016).

D     Crystal structure of human Notch2 (EGF11-13), and comparison with modified human Notch1 EGF11-13 (PDB ID = 4D0E) (Taylor *et al*, 2014) Phe-478 and Phe-516 are highlighted in Notch2 as these appear to be shielded from the solvent by the glycans on Ser-462 and Ser-500, respectively. Throughout all figures, where space does not allow full naming, ligands and Notch receptors are identified by initial letter and number, for example Notch1—N1 or Jagged1—J1.

**Human Notch2 EGF11-13 structure is highly homologous to Notch1**

We have also solved the structure of human Notch2 EGF11-13, which includes the ligand-binding region (EGF11-12) (Fig 1D and Table 1). Human Notch2 EGF11-13 has 67% sequence identity with human Notch1, and superposition of the two structures shows that both adopt a very similar linear arrangement (RMSD = 1.55 Å) (Fig 1D). In our crystal structure, Notch2 is *O*-fucosylated on

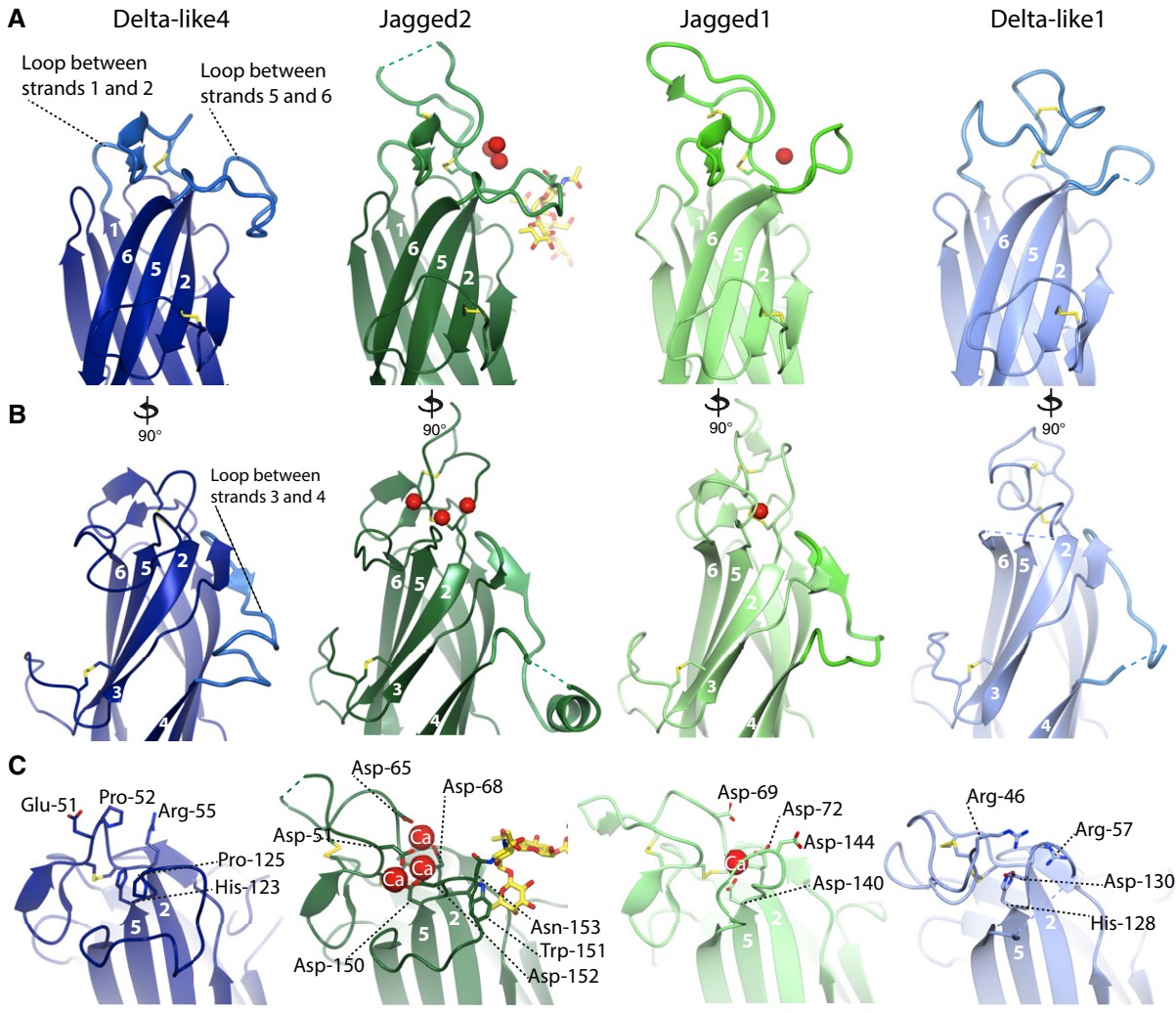

**Figure 2. Comparison of the N-terminal C2 domains in the different Notch ligands.**

A  The loops between strands 1 and 2 (calcium-binding region 1 (CBR1)), and between 5 and 6 (CBR3) of the Notch ligand C2 domains differ in length and conformation.
B  The loop between strands 3 and 4 (CBR2) is also different.
C  The Jagged1 and Jagged2 C2 domains bind calcium ions (shown in red), whereas Delta-like4 and Delta-like1 do not; the aspartates involved in calcium binding are not conserved in Delta-like4 and Delta-like1. Both Jagged1 and Jagged2 contain *N*-glycosylation sites in the loop between strands 5 and 6. The α(1,6)-linked fucose on the Asn-153 *N*-glycosylation site in Jagged2 packs against Trp-151 side chain (shown). All of these differences at the putative lipid-binding site likely reflects the different Notch ligand lipid-binding specificity.

Thr-470, and *O*-glycosylated on Ser-462 and Ser-500. In the earlier Notch1-Delta-like4 structure, the *O*-fucose on the equivalent threonine residue in EGF12 is seen to be involved in direct intermolecular binding to the C2 domain (Taylor *et al*, 2014; Luca *et al*, 2015). In contrast, the *O*-glucose modifications affecting specific serine residues in Notch2 appear to stabilize intramolecular structure as observed previously for the Notch1 complex (Luca *et al*, 2015): the disaccharide (glucose + xylose) on Ser-462 may stabilize Notch2 EGF12 through interaction with Phe-478 (shown), shielding it from the solvent. Similarly, the *O*-glucose on Ser-500 may stabilize EGF13 through interaction with Phe-516 (Fig 1D).

The availability of our new structures of hJagged2, hDelta-like4 and hNotch2 allows modelling of the receptor-binding interface across all canonical ligands and Notch1/Notch2 and show that almost all of the key residues involved in ligand binding (Leu-468, Asp-469 and Ile-477 in site 1; Phe-436 and Arg-448 in site 2) are conserved in Notch2. Receptor-binding residues in site 1 and 2 are also highly conserved across the ligands suggesting that other forms of regulation must be present to drive specific ligand–receptor pairing and signalling, rather than intrinsic differences in affinity between the core recognition elements of the various receptor–ligand pairs (Figs 3 and EV1). This is supported by assaying different combinations of receptor/ligands which show similar levels of binding (Fig 4A). Fringe extension of *O*-fucosylated sites in the receptor is already known to change the responsiveness to ligand binding, but the variation in loop sequences in C2 domains of ligands offers another potential method to fine tune the affinity of the Notch complex and subsequent signalling capability.

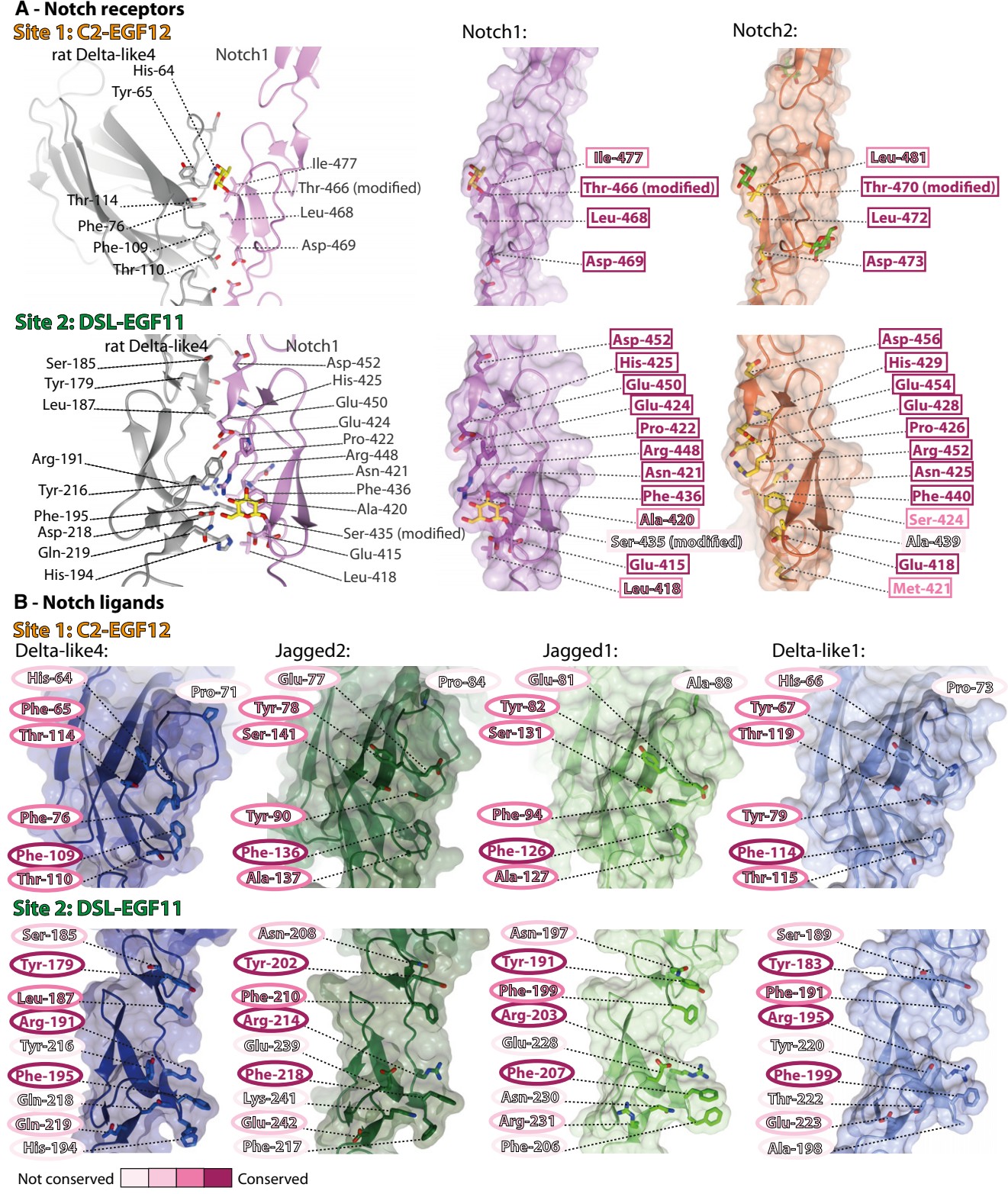

**Figure 3.  Comparison of the residues involved in complex formation in the different Notch receptors and Notch ligands.**

A, B   Comparison of the residues involved in complex formation (Luca *et al*, 2015) in Notch1 and Notch2 (A), and in the different human ligands (B), at site 1 (C2:EGF12) and site 2 (DSL:EGF11). N.b. The complex structure between Delta-like4 and Notch1 was of rat Delta-like4 (shown in panel A); human Delta-like4 is shown in panel (B). Conservation of the residues involved in complex formation is indicated by background colour, highlighting the high conservation of the ligand-binding sites in Notch1 and Notch2. There are a few residues in the receptor-binding sites of the ligands that are absolutely conserved, with site 2 (DSL:EGF11) being more variable than site 1 (C2:EGF12).

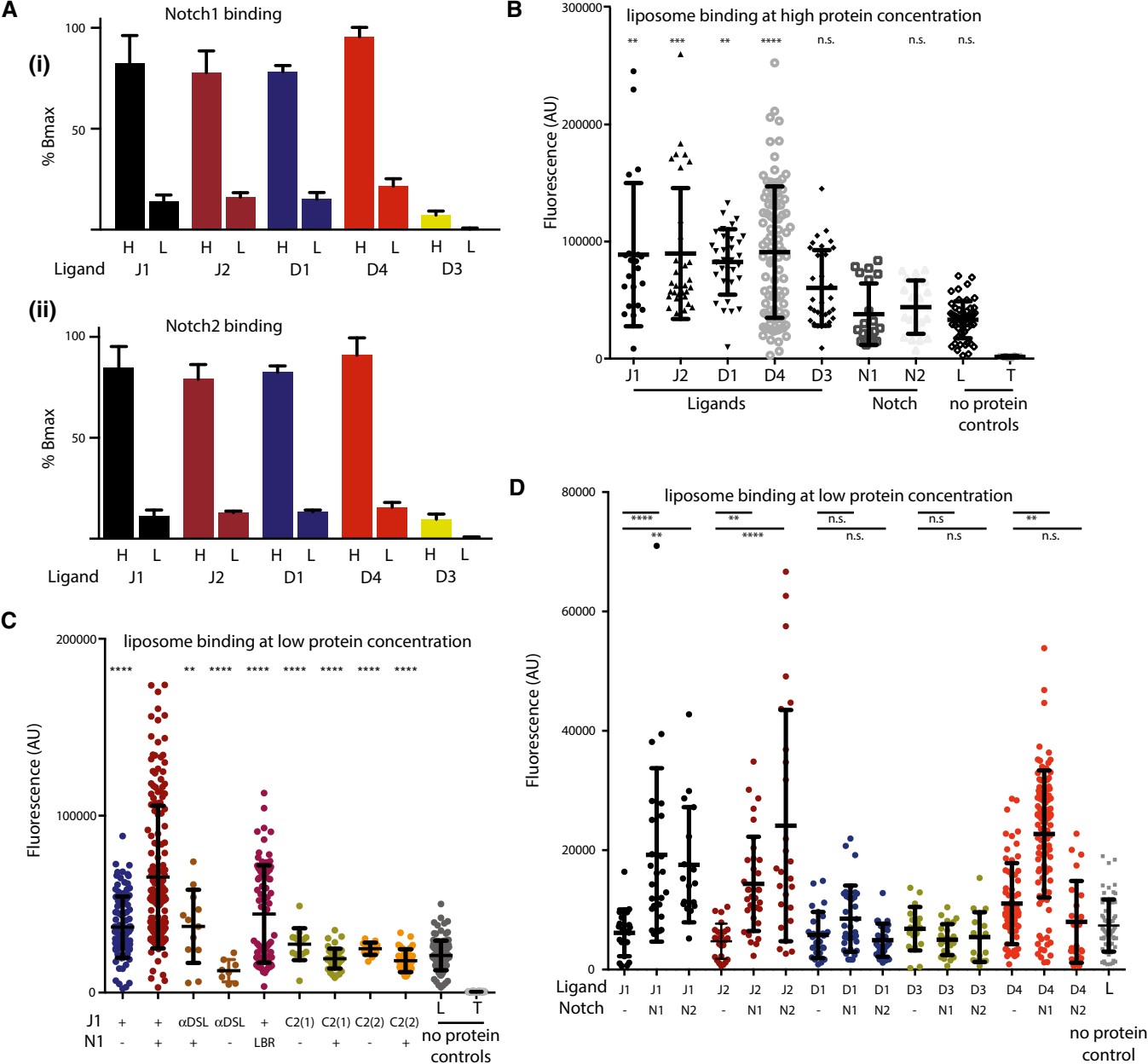

**Figure 4. Functional coupling between liposome and Notch binding of Notch ligands.**

A  All canonical human Notch ligands (Jagged1, Jagged2, Delta-like1 and Delta-like4) (N-EGF3) bind to both human Notch1 (i) and Notch2 (ii) EGF11-13 to a similar extent as assessed by plate assay. Notch ligands were bound to nickel-coated plates before biotinylated pre-clustered Notch was added with NeutrAvidin-conjugated HRP. Binding is shown at high (H, 300 nM) and low (L, 20 nM) protein concentrations with Delta-like3, which does not bind Notch, acting as a negative control at these concentrations. All components were purified from insect cells, and three independent experiments were performed with all points in duplicate in each.

B  All canonical human Notch ligands (Jagged1 (J1), Jagged2 (J2), Delta-like1 (D1) and Delta-like4 (D4)) (N-EGF3) bound to fluorescently labelled liposomes (PC/PE/PS)—Delta-like3 (N-EGF1), Notch1 (EGF11-13) and Notch2 (EGF11-13) did not bind. Four independent experiments with a minimum of 36 replicates were performed.

C  At low concentrations of ligand, that is, below concentrations where liposome binding can be observed, addition of Notch1 EGF11-13 stimulated binding of Jagged1 N-EGF3 to liposomes. This effect could be abolished by an antibody against the Notch-binding DSL domain of Jagged1 (α-DSL), or by substitution of residues critical for ligand binding to Notch (Leu468Ala) (LBR). Liposome binding was also abolished by substitutions that directly perturb the putative lipid-binding site in the C2 domain of Jagged1 (Asp140Ala/Asp144Ala, C2(1) and Del1Del2Asp140Ala, C2(2)) (Chillakuri *et al*, 2013). Fifteen independent experiments with a minimum of 18 replicates were performed.

D  Addition of Notch1 EGF11-13 stimulated binding of Jagged1 and Delta-like4 N-EGF3 to liposomes, with Notch2 having a similar effect on Jagged2, but neither had an effect on Delta-like1 or Delta-like3 in terms of liposome binding. Five independent experiments with 50 replicates were performed. Data were analysed with Prism 6 or 7 (GraphPad, San Diego, CA, USA).

Data information: Comparisons between two groups were performed with a two-tailed unpaired *t*-test. Statistical differences among various groups were assessed with ordinary one-way ANOVA by comparison to the mean of a control column. Values are presented together with the mean ± SD. **$P < 0.01$; ***$P < 0.001$; ***$P < 0.0001$.

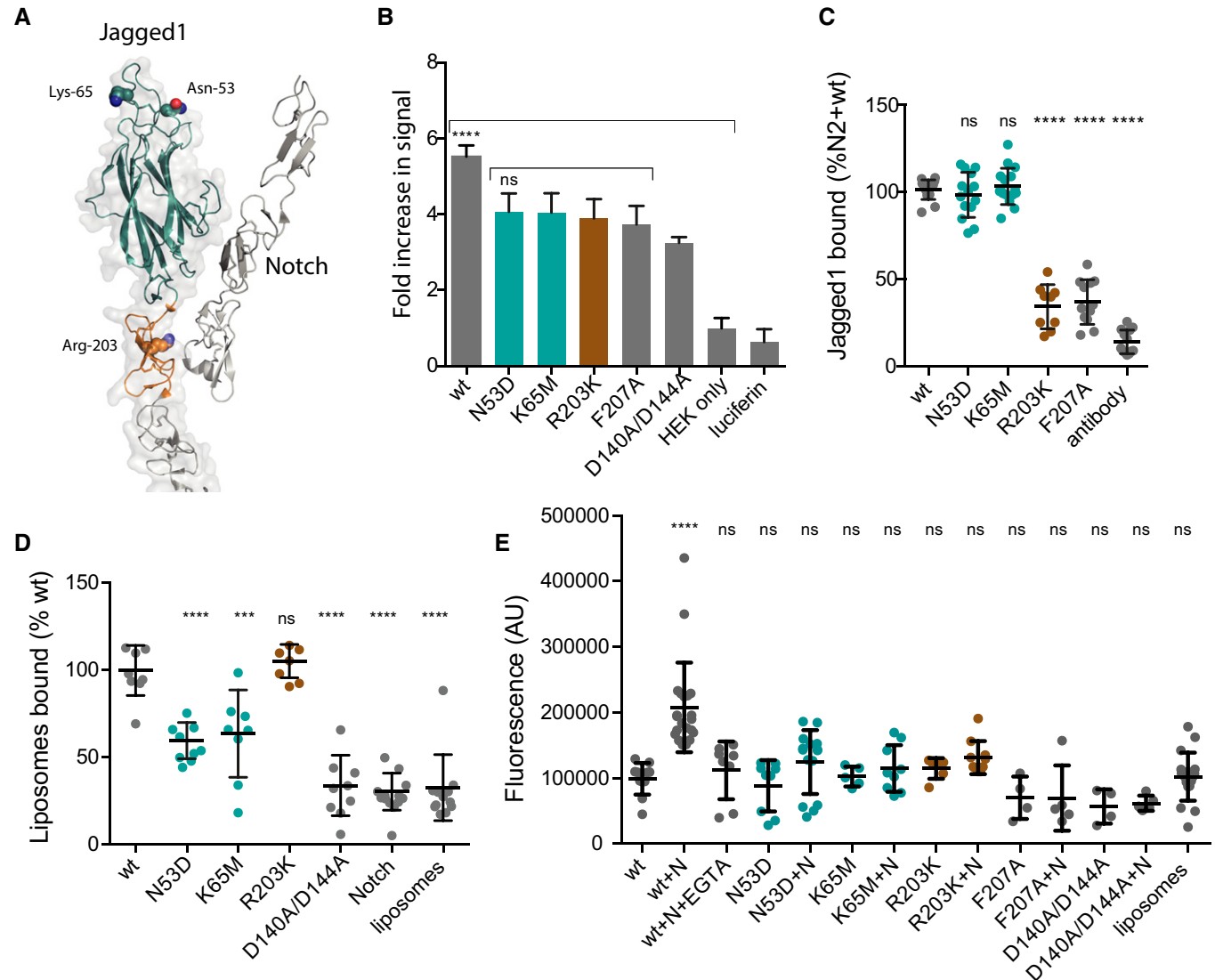

**Figure 5.   Disease-causing substitutions affecting C2 loops selectively alter membrane but not Notch binding.**

A    C2 domain of Jagged1 showing position of extrahepatic biliary atresia (EHBA) causing substitutions Asn53Asp and Lys65Met in loop regions, the position of Arg203Lys in DSL Notch-binding site also associated with EHBA is shown.

B–E  C2 EHBA variants reduce Notch1 (data not shown) and Notch2 activation (B) C2 EHBA variants do not affect Notch binding, unlike Arg203Lys (C) but liposome binding is reduced and the Notch boosting effect is lost (D, E). At least three independent experiments with ten replicates were performed for all assays. Data were analysed with Prism 6 or 7 (GraphPad, San Diego, CA, USA). Comparisons between two groups were performed with a two-tailed unpaired $t$-test. Statistical differences among various groups were assessed with ordinary one-way ANOVA by comparison to the mean of a control column. Values are presented together with the mean ± SD. ***$P < 0.001$; ***$P < 0.0001$.

### Crosstalk between Notch and liposome binding

Purification of N-terminal fragments of all of the human Notch ligands allowed us to assess lipid binding of these constructs. All of the canonical Notch ligands (Jagged1, Jagged2, Delta-like1 and Delta-like4) bound to liposomes consisting of a mixture of phosphatidylcholine/phosphatidylserine/phosphatidylethanolamine (PC/PS/PE); however, Delta-like3 (non-canonical) did not show significant binding (Fig 4B). As noted above, the putative lipid-binding site at the extreme termini of the Notch ligands is a site of considerable sequence diversity between the various ligands which

we hypothesize is likely to confer different lipid-binding specificities to each. To test this, we investigated a range of other liposome lipid compositions but many were not compatible with this assay; however, we could demonstrate preferences in binding between different ligands using ganglioside- or sphingomyelin-rich liposomes (Fig EV2). It is interesting to note that prior studies have implicated, at a genetic level, glycosphingolipids as being important in Notch signalling in both flies and worms (Hamel *et al*, 2010; Pontier & Schweisguth, 2012; Katic & Greenwald, 2015), although these approaches could not differentiate between direct and indirect involvement of glycosphingolipids in the signalling pathway. To

investigate whether Notch ligands are able to bind to both lipids and Notch simultaneously, we added Notch1 EGF11-13 into our liposome assays. When working at low concentrations of ligand, that is, below concentrations where liposome binding can be observed, the addition of Notch1 stimulated liposome binding to Jagged1 (Fig 4C). This stimulation is abolished when residues in the C2 domain involved in calcium binding were mutated [D140A/D144A, C2(1)], or when the two loops forming the putative lipid-binding site in the C2 domain were substantially shortened [Del1Del2D140A, C2(2)]. In addition, liposome binding was also abolished when Notch binding was inhibited by either inclusion of an antibody recognizing the receptor-binding site of the DSL domain of Jagged1 (α-DSL), or use of a Notch1 variant L468A, defective in ligand binding (LBR) (Fig 4C).

To investigate whether or not this stimulation of liposome binding is seen for all Notch ligands, we set up analogous liposome-binding assays adding either Notch1 or Notch2 EGF11-13. Addition of Notch1 to liposome assays enhanced binding of Jagged1, Jagged2 and Delta-like4 to liposomes, whilst Notch2 enhanced binding to Jagged1, Jagged2 but not Delta-like4 (Fig 4D). No enhancement is seen with either Notch1 or Notch2 for Delta-like1 or Delta-like3 liposome binding. The enhancement of liposome binding seen for some of the ligands upon addition of Notch may be due to Notch binding to and rigidifying the ligands, and thereby decreasing the loss of entropy upon binding to the lipids/membrane. Such coupling between the two binding events gives a mechanism to increase the affinity of the receptor/ligand complex and enhance signalling by a specific ligand in a particular physiological context and/or to affect selection by Notch of one ligand from a pool of ligands expressed on the cell surface, despite the apparently similar affinities of each Notch for all ligands. Thus, the lipid composition of the cell membrane could act as a modulator of Notch signalling, in addition to *O*-glycosylation of the receptor.

### Human disease-associated variants alter membrane but not Notch binding

Taken together, our data strongly support a key role for the ligand C2 domains in Notch signalling independent of direct Notch engagement. Point mutations in *Jagged1* have previously been linked with two diseases: Alagille syndrome (Penton *et al*, 2012) and extrahepatic biliary atresia (Kohsaka *et al*, 2002). Alagille is a more severe multi-system disease, and we have previously demonstrated that the amino acid substitutions linked to this disease result in misfolded protein being retained within the cell and hence lead to haploinsufficiency of Jagged1 (Chillakuri *et al*, 2013). The *Jagged1* point mutations in EHBA affect only one organ system and lead to an abnormal or absent extrahepatic bile duct implying a more subtle effect on Notch signalling. We therefore generated recombinant Jagged1 variant proteins bearing three of the individual amino acid substitutions associated with EHBA—Asn53Asp and Lys65Met within the apical loops of the C2 domain and Arg203Lys within the Notch-binding interface of the DSL (Fig 5A). All these variant forms could be purified as recombinant proteins unlike those previously studied Alagille variants, suggesting that haploinsufficiency of Jagged1 does not cause EHBA. All variants reduced activation in a cellular assay of Notch signalling activity to a similar level to control substitutions (Phe207Ala within the Notch-binding site or

Asp140Ala/Asp144Ala within the C2 domain; Fig 5B). To define the mechanisms leading to reduction in activity, we next tested the variant proteins to see whether either membrane or Notch recognition was altered. As predicted, since it involves replacing the larger Arg with a shorter Lys, the disease-causing variant within the Notch-binding site abrogated Notch binding (Fig 5C) but left liposome binding intact (Fig 5D). Conversely, both disease-causing C2 domain variants left Notch binding unaltered (Fig 5C), but significantly reduced liposome binding (Fig 5D). All three disease-causing variants therefore did not show the enhancement of liposome binding in the presence of Notch seen for WT Jagged1 (Fig 5E). These data strongly suggest that EHBA is caused by a failure to form a ternary complex comprising membrane, Notch and ligand, which in the cases studied here can be due to a reduction in membrane recognition by the C2 domain (Asn53Asp, Lys65Met) or a direct effect on Notch recognition mediated by the DSL domain (Arg203Lys). These data therefore strongly support the idea that two recognition events are critical for efficient Notch signalling in some physiological contexts, such as bile duct development.

Since composition of membrane can vary with cell type and stage of development, future studies will investigate the *in vivo* importance of the ternary complex for Notch signalling utilizing CRISPR-Cas genome editing approaches in combination with structure-informed mutagenesis of C2 domain loops from different Notch ligands.

## Materials and Methods

Full experimental details are provided in the Appendix.

### Protein expression and production

Notch ligand and receptor constructs were recombinantly expressed in S2 insect cells (Expres2ion® Biotechnologies, Denmark) as C-terminally His-tagged fusion proteins. Media containing recombinantly expressed protein were filtered and loaded onto a cOmplete His-tag Purification Column (Roche Diagnostics, UK), for purification via His-tag. Following washing with 50 mM Tris pH 9.0, 5 mM imidazole pH 8.0, 200 mM NaCl and 1 mM CaCl$_2$, proteins were eluted with buffer containing 250 mM imidazole pH 8.0. Following overnight dialysis, proteins were further purified by size-exclusion chromatography (SEC) using a Superdex S200 (ligands) or S75 (receptors) preparative column in 20 mM Tris pH 7.5, 200 mM NaCl and 1 mM CaCl$_2$.

### Structure determination of extracellular fragments of human Notch ligands and the Notch2 receptor

Delta-like4 N-EGF3 was crystallized at 3 mg/ml in 0.1 M MES pH 6.5, 12% (w/v) PEG-20K at a 3:1 protein:precipitant ratio. Crystals were cryoprotected with 35% (v/v) glycerol, and data were collected to 2.2 Å (Table 1). The crystal belonged to space group C2, with one molecule in the asymmetric unit. The structure was solved by molecular replacement using the C2 domain of Jagged1 (PDB ID = 4CC1) (Chillakuri *et al*, 2013) using *Phaser* (McCoy *et al*, 2007) within *CCP4* (Winn *et al*, 2011), before the remaining domains were found also using Jagged1 as a search model. The

structure was built using the automatic model building software *Buccaneer* (Cowtan, 2006). All structures were refined using *Coot* (Emsley *et al*, 2010), *Refmac* (Murshudov *et al*, 1997) and *Phenix refine* (Afonine *et al*, 2012).

Jagged2 N-EGF3 was crystallized at 2.3 mg/ml in the presence of 10 mM $BaCl_2$ in 0.1 M sodium citrate pH 5.3, 20% (w/v) PEG-5000 MME at a 3:1 protein:precipitant ratio. Crystals were cryoprotected with 30% (v/v) ethylene glycol, and data were collected to 2.8 Å (Table 1). The crystal belonged to space group $P2_12_12_1$, with one molecule in the asymmetric unit. The structure was solved by molecular replacement using the C2 domain of Jagged1 (PDB ID = 4CC1) (Chillakuri *et al*, 2013) using *Phaser* (McCoy *et al*, 2007), before the remaining domains of Jagged1 N-EGF3 were placed sequentially into the electron density with iterative rounds of rigid body and restrained refinement in *Refmac* (Murshudov *et al*, 1997). Barium was not visible in the electron density maps, and neither were the loops at the tip of the C2 domain.

Jagged2 N-EGF2 was crystallized at 3.4 mg/ml in the presence of 10 mM $CaCl_2$ in 0.05 M potassium dihydrogen phosphate, 20% (w/v) PEG-8000 at a 3:1 protein:precipitant ratio. Crystals were cryoprotected with 20% (v/v) glycerol, and data were collected to 2.7 Å (Table 1). The crystal belonged to space group $P2_12_12_1$ (similar to the apo N-EGF3 crystal form), with one molecule in the asymmetric unit. The structure was solved by molecular replacement using the N-EGF2 portion of the apo N-EGF3 model using *Phaser* (McCoy *et al*, 2007). Three putative calcium ions and most of the residues in the loops of the C2 domain are visible in the electron density. Calcium ions were assigned on the basis of (i) their anomalous scattering, (ii) electron density peak height c.f. the protein atoms providing the metal ligands, (iii) the nature of the protein ligands and (iv) their appropriate refinement (in terms of B factors c.f. ligand atoms at full occupancy). Overlays with other C2 domains that reveal the coincidence in location of calcium-binding sites were not performed until refinement had converged (see main text).

Jagged2 N-EGF2 also crystallized at 3.3 mg/ml in the presence of 20 mM $CaCl_2$ in 0.1 M sodium cacodylate pH 6.5, 0.2 M ammonium sulphate, 30% (w/v) PEG-8000 at a 1:1 protein:precipitant ratio. Crystals were cryoprotected with 20% (v/v) ethylene glycol, and data were collected to 2.8 Å (Table 1). The crystal belonged to space group C2, with six molecules in the asymmetric unit. The structure was solved by molecular replacement using the N-EGF1 portion of the above N-EGF2 structure model using *Phaser* (McCoy *et al*, 2007), before the EGF2 domains were placed manually into the electron density. Three putative calcium ions were assigned (defined as described above) bound to the C2 domain, with the loops of the C2 domain mostly visible (excluding three residues in the loop between strands 1 and 2). Asn-153 is *N*-glycosylated with density representing the first four sugar moieties visible (until the β (1–4)-linked mannose).

Notch2 EGF11-13 crystallized at 20 mg/ml in the presence of 10 mM $CaCl_2$ in 0.1 M sodium cacodylate pH 6.5, 0.2 M sodium acetate, 30% (w/v) PEG-8000 at a 3:1 protein:precipitant ratio. Crystals were cryoprotected with 15% (v/v) ethylene glycol, and data were collected to 1.9 Å (Table 1). The crystal belonged to space group $P2_12_12_1$ with one molecule in the asymmetric unit. The structure was solved by molecular replacement using the individual EGF domains of Notch1 (PDB ID = 2VJ3) (Cordle *et al*, 2008). EGF12

was found first, before EGF11 and EGF13. Density representing *O*-glucose on Ser-462 and Ser-500, and *O*-fucose on Thr-470 was clearly visible. Density representing xylose linked to the *O*-glucose on Ser-462 was also visible.

## Determination of apparent binding affinities by plate assay

Pierce 96-well nickel-coated plates (Thermo Scientific) were coated with Notch ligands (5 μg/ml) (N-EGF3 constructs) in 20 mM HEPES pH 7.4 containing 200 mM NaCl (HBS). After incubation at 4°C overnight, the plate was washed and blocked for 1 h at room temperature with 4% (w/v) bovine serum albumin in the same buffer. Human N1 and N2 EGF11-13 constructs (300 nM) that had been biotinylated by site-specific biotin ligase (Avidity) were mixed in a 4:1 molar ratio with NeutrAvidin-HRP conjugate (Life Technologies) at room temperature for 1 h in HBS containing 5 mM $CaCl_2$. The pre-clustered Notch constructs at a range of concentrations from 300 nM (high concentration) and 20 nM (low concentration) were added to the ligand-coated plates and incubated for 1.5 h at room temperature. After washing four times with HBS containing 5 mM $CaCl_2$ and 0.05% (v/v) Tween-20, followed by two washes in the absence of Tween-20, the plate was developed with 2,2′-Azino-bis(3-ethylbenzothiazoline-6-sulphonic acid) diammonium salt (Sigma-Aldrich). Absorbance was measured with a PHERAstar FS microplate reader (BMG LABTECH). Notch binding at the high (H) and low (L) concentrations is shown in Fig 4A.

## Liposome-binding assays

Notch ligands (e.g. Jagged1 N-EGF3) were coated at 800 nM/200 nM in a 40 μl volume onto a nickel-coated plate (Pierce Nickel-Coated Plates, black, 96-well, #15342) in 20 mM HEPES pH 7.4, 200 mM NaCl through incubation overnight at 4°C. The former concentration was used for testing liposome binding of the Notch ligands, and the latter for incorporating hNotch1 EGF11-13 into the assays. Controls used to demonstrate specific binding were liposomes (L) added in the absence of ligand and Triton X-100 (T) added in the presence of ligand, but absence of liposomes. Both were subsequently blocked as for experimental samples.

Wells were washed twice with 20 mM HEPES pH 7.4, 200 mM NaCl before blocking for ~3 h with 0.1% (w/v) gelatin in the same buffer. Following incubation, wells were washed twice more with buffer, with the second wash buffer also containing 5 mM $CaCl_2$.

To appropriate wells, 8 μl of anti-Jagged1 DSL domain Ab supernatant was added.

Liposomes were prepared as described in Chillakuri *et al* (Chillakuri *et al*, 2013). To analyse the effect of adding Notch1 EGF11-13 on liposome binding, Notch1 EGF11-13 was added at a 4:5 liposome:protein volume ratio, with liposomes diluted 1 in 20 in 20 mM HEPES pH 7.4, 200 mM NaCl, 5 mM $CaCl_2$ and Notch1 at 800 nM in the same buffer.

The liposome:protein mix was incubated on a rotating shaker for 1 h, before washing twice in 20 mM HEPES pH 7.4, 200 mM NaCl, 5 mM $CaCl_2$. Liposomes were solubilized in 40 μl of 0.3% Triton X-100 for at least 1 h before the fluorescence intensity was measured in 96-well plate reader format using PHERAstar BMG Labtech (wavelengths: excitation 485 nm, emission 520 nm).

### Assay of EHBA variant proteins

Ligand proteins were produced as Fc fusions in human embryonic kidney 293T (HEK293T) cells using a transient transfection system (Aricescu *et al*, 2006) and purified as described in Chillakuri *et al*, 2013 (Chillakuri *et al*, 2013). Notch1/2 EGF11-13 constructs were produced in S2 system and purified as above. Notch activation assays were performed as in Chillakuri *et al*, 2013 with Notch1 (data not shown) and Notch2 reporter cell lines (kind gift from R. Kopan) (Liu *et al*, 2013). Notch binding to EHBA variants was performed according to the following: briefly, 200 ng (50 μl volume) of monomeric Notch1/2 EGF11-13 construct in 20 mM HEPES pH 7.4, 200 mM NaCl buffer was immobilized on a transparent 96-well Maxisorp immuno-plate overnight at 4°C. The wells were then washed three times with 200 μl buffer and blocked with 200 μl of blocking buffer: 20 mM HEPES pH 7.4, 200 mM NaCl, 3% milk, 0.1% gelatin, 5 mM CaCl$_2$. After 90 min (25°C), wells were washed and a 50 μl solution of 200 nM dimeric ligands constructs was added. 90 min later, wells were washed and incubated with a 1:2,000 dilution of mouse–anti-human IgG Fc antibody conjugated to HRP for 60 min. After washing, 100 μl of 2,2′-Azino-bis(3-ethyl-benzothiazoline-6-sulphonic acid) diammonium salt (Sigma-Aldrich) substrate solution was added to each well. The absorbance of the plate was read at a wavelength of 415 nm using a PHERAstar plate reader.

Liposome-binding experiments were performed as above except black plates were coated with 40 μl of 800 nM ligands in 20 mM HEPES pH 7.4, 200 mM NaCl. To determine whether Notch increased liposome binding to EHBA variants 40 μl of 200 nM dimeric ligands were coated overnight at 4°C onto a black plate in 20 mM HEPES pH 7.4, 200 mM NaCl and the assay performed as described above.

### Statistical analyses

All data were analysed with Prism 6 or 7 (GraphPad, San Diego, CA, USA). Comparisons between two groups were performed with a two-tailed unpaired *t*-test. Statistical differences among various groups were assessed with ordinary one-way ANOVA by comparison to the mean of a control column. Values are presented together with the mean ± SD.

**Expanded View** for this article is available online.

### Acknowledgements

RJS and BK were funded by a Medical Research Council Grant MR/L001187/1 & CM by Wellcome Trust Grant 097928 to PAH and SML. SML is supported by a Wellcome Trust Investigator Award 100298. PW is supported by a grant awarded by the EPA Cephalosporin Fund. We thank Diamond Light Source for access to beamlines I02, I03, I04 and I04-1 (MX12346) that contributed to the results presented here. All protein structures and X-ray data have been deposited in the Protein Data Bank with the identifiers shown in Table 1.

### Author contributions

RJS and BK generated protein reagents with assistance in initial derivation of some cell lines from CC and LH. RJS performed crystallization trials, and RJS and SML collected X-ray data, solved and analysed atomic structures. BK, PW and PAH designed all other assays. BK performed all liposome-binding assays, and PW performed Notch-binding assays. BK and SML performed statistical analyses of assays. RJS, PAH and SML wrote a first draft of the manuscript to which all other authors contributed.

### Conflict of interest

The authors declare that they have no conflict of interest.

### Note added in proof

The structure of a Jagged1/Notch1 EGF8-12 complex has been solved (Luca *et al*, 2017). The authors note the rearrangement of ligand we predict.

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
