## [Review Process File · The EMBO Journal]

Manuscript EMBO-2017-96632

Structural and functional dissection of the interplay between lipid and Notch binding by human Notch ligands

Richard J. Suckling, Boguslaw Korona, Pat Whiteman, Chandramouli Chillakuri, Laurie Holt, Penny A. Handford & Susan M. Lea

Corresponding authors: Penny A. Handford, University of Oxford and Susan M. Lea, Sir William Dunn School of Pathology

Review timeline:	Submission date:	30 January 2017
	Editorial Decision:	07 March 2017
	Revision received:	16 March 2017
	Editorial Decision:	30 March 2017
	Revision received:	06 April 2017
	Accepted:	07 April 2017

Editor: Ieva Gailite

Transaction Report:

1st Editorial Decision	07 March 2017
---------------

Thank you for submitting your manuscript for consideration by the EMBO Journal. We have now received two referee reports on your manuscript, which are included below for your information.

Based on the referees' comments and the information you provided in the pre-decision discussion, I would like to invite you to submit a revised version of the manuscript, addressing the comments of both reviewers. Particularly I would like to ask you to focus on the following points:

- Include additional information on specificity of lipid binding by Notch ligands (referee #1)
- Include negative controls for Notch/ligand binding experiments shown in figure 4A (referee #2)
- Provide further information on calcium ion binding by Jagged2, as requested by both referees

I should add that it is The EMBO Journal policy to allow only a single major round of revision and that it is therefore important to resolve the main concerns at this stage.

When preparing your letter of response to the referees' comments, please bear in mind that this will form part of the Review Process File, and will therefore be available online to the community. For more details on our Transparent Editorial Process, please visit our website: http://emboj.embopress.org/about#Transparent_Process

We generally allow three months as standard revision time, but an extension to six months is possible in case of extensive revisions. Please contact us in advance if you need an extension. As a

matter of policy, competing manuscripts published during this period will not negatively impact on our assessment of the conceptual advance presented by your study. However, we request that you contact the editor as soon as possible upon publication of any related work to discuss how to proceed.

Please feel free to contact me with any further questions regarding the revision. Thank you for the opportunity to consider your work for publication. I look forward to your revision.

REFEREE REPORTS

Referee #1:

In this paper, S. Lea and P Handford present the structures of human Jagged2 and DLL4 (more precisely, of N-terminal fragments covering the C2 and DSL domains + the EGF-like repeats 1-3). Analysis of the Jagged2 structure led the authors to suggest that the two different conformations reported so far for the Notch ligands likely result from crystal packing (Figure 1). Also, the comparison between the proposed structure of DLL4 and Jagged2 (this study) with Jagged1 and DLL1 (reported earlier) indicated structural differences in the loops bulging out of the C2 domain (Figure 2). This raises the possibility that different ligands recognize different lipids at the cell surface (and possibly organize lipid nanoclusters).

This study also reports on the structure of a human Notch2 fragment implicated in ligand binding (EGF11-13), which appeared quite similar to the structure of the corresponding region of human Notch1 (Figure 3). This piece of information does not appear to be key for this study (hence could be moved to the Sup Data section to improve the logical flow).

To test the potential role of lipid-C2 domain interaction in Notch signaling, the authors first performed a liposome-interaction assay and showed that Jagged1, Jagged2, DLL1 and DLL4 N-terminal fragments interact *in vitro* with liposomes (with a given PC/PS/PE composition). The addition of Notch1/Notch2 enhanced some of these ligand-lipid interaction, indicating that ligand-receptor interactions contribute to stabilize C2-lipid interactions (Figure 4).

The authors then examined the *in vitro* properties of human disease variants that are presumably associated with only mild reduction in Jagged1 activity. They found that a mutation in the Notch-binding site interfered with ligand-receptor interaction but not with ligand-liposomes interactions; whereas mutations in the loops of the C2 domain affected liposome binding but not receptor binding (Figure 5). These data suggest that lipid binding modulates the *in vivo* activity of Jagged2.

This is a short and straightforward structural paper. The conclusions are well supported by the data. This study is interesting in that it raises several questions. These remain, however, unaddressed. For instance: What are the lipids bound by the different ligands? Do different ligands recognize different lipids? If so, how do the C2 loops contribute to lipid binding specificity? What would be the functional significance of ligand-specific interaction with lipids? Can lipid-binding specificity be transferred from one ligand to another? do ligands interact with cell surface lipids *in cis* or *in trans*? Do ligands organize lipid nanodomains in signal sending/receiving cells? Any attempt to identify the lipids bound by these different C2 domains and to test binding specificity would clearly make this paper of broader interest.

Minor points

- p. 4: 'RMSD', 'DALI search': please define
- p. 4: 'the C2 loop of jagged2 is ordered upon calcium binding/ three calcium ions are bound in the Jagged2 C2 domain': please justify these statements (or provide references)
- p. 6: 'thus the lipid composition of the cell membrane could act as a modulator of Notch signaling': there are a few studies that have linked Notch signaling to membrane lipid composition. These could be cited.

Referee #2:

The manuscript submitted by Suckling et al provides additional structure-function insights into interactions between Notch receptors and ligands, and ligands interacting with phospholipids via their C2 domains. Noteworthy findings include new structures of the ligands Jagged2 and DLL4, and the ligand binding region of the Notch2 receptor; characterization of the liposome binding properties of ligands; and interestingly, putative coupling between ligand-liposome interactions with Notch-ligand binding. Overall, the manuscript is clear and concise, and makes significant contributions to our overall understanding of how Notch receptors and ligands interact in order to initiate signaling. If the authors were to address my relatively minor concerns summarized below, then I would recommend that their revised manuscript be published in the EMBO journal.

Comments (in no particular order):

- Are any of the EGF repeats in the ligand structures predicted to bind calcium? It seems that previous work by the authors and others suggest that calcium binding leads to rigid linear EGF repeats and repeats that don't bind calcium often are either flexible or have distinct bend angles. Can any structural insights be drawn from this with regards to the linearity/bends of the ligand structures?
- What were the criteria used for assigning Ca²⁺ ions in the Jagged2 structures? This might be useful to include in the methods section.
- From the methods section and main body of the manuscript, it is not clear to me whether calcium is always included in the liposome binding assays. For the Jagged ligands, which seem to bind calcium, does this make a difference on liposome interactions if calcium is excluded in these assays?
- Giving the clinical relevance of mutations in the C2 domain of Jagged1 and EHBA, if the authors did their cellular signaling assays (Fig 5B) in a hepatic cell line do they think the mutations might have a stronger effect in these reporter assays? Really a comment more than a question.
- In the authors' receptor-ligand plate binding assay, the receptors are pre-clustered via neutravidin. Is pre-clustering required in order to see binding in this assay?
- It is not clear from Figure 4A what binding to low and high protein concentrations refers to, ligand or receptor. The methods mention 0-300nM Notch is used in the assays. Can the authors clarify how this assay is performed?
- Also regarding Figure 4A, there are no negative controls, e.g. a ligand binding mutant or an EGF construct not involved in ligand-receptor complexes, and because of this, it is not clear to me whether binding is observed at the "low" protein concentrations or not.
- Is there a mistake in the asterisks in Figure 4C that denote statistical significance? I would have thought that the pairwise comparisons should be with the first lane (J1+/N1+). Is this correct?
- I find the binding result for the EHBA mutation R203K rather striking (Figure 5C). Could the authors include a structural basis for this seemingly conservative change?

1st Revision - authors' response

16 March 2017

Please find below a detailed response to the referees including new data incorporated and changes made to the manuscript. We hope that this version of our manuscript will prove acceptable to EMBO J.

Referee #1:

In this paper, S. Lea and P Handford present the structures of human Jagged2 and DLL4 (more precisely, of N-terminal fragments covering the C2 and DSL domains + the EGF-like repeats 1-3). Analysis of the Jagged2 structure led the authors to suggest that the two different conformations reported so far for the Notch ligands likely result from crystal packing (Figure 1). Also, the comparison between the proposed structure of DLL4 and Jagged2 (this study) with Jagged1 and DLL1 (reported earlier) indicated structural differences in the loops bulging out of the C2 domain (Figure 2). This raises the possibility that different ligands recognize different lipids at the cell surface (and possibly organize lipid nanoclusters).

This study also reports on the structure of a human Notch2 fragment implicated in ligand binding (EGF11-13), which appeared quite similar to the structure of the corresponding region of human Notch1 (Figure 3). This piece of information does not appear to be key for this study (hence could be moved to the Sup Data section to improve the logical flow).

We acknowledge the referees querying of the logical flow of the paper, but would prefer to keep the structure of the other, key human Notch fragment, used in our binding assays in the main text.

To test the potential role of lipid-C2 domain interaction in Notch signaling, the authors first performed a liposome-interaction assay and showed that Jagged1, Jagged2, DLL1 and DLL4 N-terminal fragments interact in vitro with liposomes (with a given PC/PS/PE composition). The addition of Notch1/Notch2 enhanced some of these ligand-lipid interaction, indicating that ligand-receptor interactions contribute to stabilize C2-lipid interactions (Figure 4).

The authors then examined the in vitro properties of human disease variants that are presumably associated with only mild reduction in Jagged1 activity. They found that a mutation in the Notch-binding site interfered with ligand-receptor interaction but not with ligand-liposomes interactions; whereas mutations in the loops of the C2 domain affected liposome binding but not receptor binding (Figure 5). These data suggest that lipid binding modulates the in vivo activity of Jagged2.

This is a short and straightforward structural paper. The conclusions are well supported by the data. This study is interesting in that it raises several questions. These remain, however, unaddressed. For instance: What are the lipids bound by the different ligands? Do different ligands recognize different lipids? If so, how do the C2 loops contribute to lipid binding specificity? What would be the functional significance of ligand-specific interaction with lipids? Can lipid-binding specificity be transferred from one ligand to another? do ligands interact with cell surface lipids in cis or in trans? Do ligands organize lipid nanodomains in signal sending/receiving cells?

We are glad the referee finds the paper interesting and agree that these are exciting questions. We feel that this manuscript goes substantially further towards elucidating the functional significance than our previous work, since it demonstrates i) that lipid and Notch-binding synergise and ii) that these properties are separable when a cohort of disease-causing variants are studied— however the rest of the referees questions are beyond the scope of this manuscript and will likely require further cellular and *in vivo* studies.

Any attempt to identify the lipids bound by these different C2 domains and to test binding specificity would clearly make this paper of broader interest.

We agree with the referee that more information regarding which specific lipids are the preferred binding partners for each ligand would be very biologically informative and we have tried many different experiments to get at these answers. To date, we cannot provide experiments that definitively identify the partners. We can, however, show some additional data (Supplemental Figure 2) that clearly support our hypothesis that the sequence variation observed in the predicted lipid-binding region of ligand C2 domains will lead to differences in the way in which they interact with lipids. The assay now presented in Supplemental Figure 2 clearly shows differential recognition of liposomes depending on the lipid composition by the different ligands. We have not included this as a main paper figure as the experimental conditions needed to successfully make liposomes with these more complex lipids means that

the liposome composition is not physiologically relevant. We have added a brief discussion of the figure to the main text:

All of the canonical Notch ligands (Jagged1, Jagged2, DLL1 and DLL4) bound to liposomes consisting of a mixture of phosphatidylcholine/phosphatidylserine/phosphatidylethanolamine (PC/PS/PE), however DLL3 (non-canonical) did not show significant binding (Figure 4B). **As noted above the putative lipid binding site at the extreme termini of the Notch ligands, is a site of considerable sequence diversity between the various ligands which we hypothesise is likely to confer different lipid binding specificities to each. To test this we investigated a range of liposome lipid compositions but many were not compatible with this assay, however we could demonstrate preferences in binding between the different ligands using ganglioside- or sphingomyelin-rich liposomes (Supplemental Figure 2).**

Minor points

- p. 4: 'RMSD', 'DALI search': please define

Defined in text on first occurrence

- p. 4: 'the C2 loop of jagged2 is ordered upon calcium binding/ three calcium ions are bound in the Jagged2 C2 domain': please justify these statements (or provide references)

The loops became ordered and the putative calcium ions were only seen following crystallization in the presence of 10-20mM CaCl₂ - the crystals grown in the presence of BaCl₂ were disordered at the tip of the protein and contained no apparent bound ions (coordinates/X-ray data for the apo-form are deposited in the PDB under accession code 5MV7 for the interested reader). Calcium ions were assigned on the basis of (1) their anomalous scattering (2) electron density peak height c.f. the protein atoms providing the metal ligands (3) the nature of the protein ligands and (4) their appropriate refinement (in terms of B factors c.f. ligand atoms at full occupancy). Overlays with other C2 domains which reveal the coincidence in location of calcium binding sites, were not performed until refinement had converged. We have added these details to the methods.

...model using Phaser (McCoy et al., 2007). Three putative calcium ions and most of the residues in the loops of the C2 domain are visible in the electron density. Calcium ions were assigned on the basis of (1) their anomalous scattering (2) electron density peak height c.f. the protein atoms providing the metal ligands (3) the nature of the protein ligands and (4) their appropriate refinement (in terms of B factors c.f. ligand atoms at full occupancy). Overlays with other C2 domains that reveal the coincidence in location of calcium-binding sites, were not performed until refinement had converged (see main text).

- p. 6: 'thus the lipid composition of the cell membrane could act as a modulator of Notch signaling': there are a few studies that have linked Notch signaling to membrane lipid composition. These could be cited.

We apologise for omission and have added a discussion point and the citations in the main text.

It is interesting to note that prior studies have implicated, at a genetic level, glycosphingolipids as being important in Notch signalling in both flies and worms (Pontier & Schweisguth, 2012; Hamel et al., 2010; Katel et al., 2005) although these approaches could not differentiate between direct or indirect involvement of glycosphingolipids in the signaling pathway.

Referee #2:

The manuscript submitted by Suckling et al provides additional structure-function insights into interactions between Notch receptors and ligands, and ligands interacting with phospholipids via their C2 domains. Noteworthy findings include new structures of the ligands Jagged2 and DLL4, and the ligand binding region of the Notch2 receptor; characterization of the liposome binding

properties of ligands; and interestingly, putative coupling between ligand-liposome interactions with Notch-ligand binding. Overall, the manuscript is clear and concise, and makes significant contributions to our overall understanding of how Notch receptors and ligands interact in order to initiate signaling. If the authors were to address my relatively minor concerns summarized below, then I would recommend that their revised manuscript be published in the EMBO journal.

We thank the referee for their positive comments about our work.

Comments (in no particular order):

• Are any of the EGF repeats in the ligand structures predicted to bind calcium? It seems that previous work by the authors and others suggest that calcium binding leads to rigid linear EGF repeats and repeats that don't bind calcium often are either flexible or have distinct bend angles. Can any structural insights be drawn from this with regards to the linearity/bends of the ligand structures?

None of the ligand EGF domains included in our structures are calcium binding. We have added a comment specifically noting this in the structure descriptions. We have inserted a sentence in the figure legend clarifying this. Since we and others (Chillakuri et al., 2013; Luca et al., 2016; Kershaw et al., 2016) have previously noted that non-calcium binding EGF domains can also pack in fairly linear, ordered, arrays we do not feel further specific comment is required here.

(Legend for Figure 1)

...the electron density). None of the EGF domains are of the calcium binding type. All of these copies...

• What were the criteria used for assigning Ca²⁺ ions in the Jagged2 structures? This might be useful to include in the methods section.

See response to referee 1 above

• From the methods section and main body of the manuscript, it is not clear to me whether calcium is always included in the liposome binding assays. For the Jagged ligands, which seem to bind calcium, does this make a difference on liposome interactions if calcium is excluded in these assays?

We apologise that this was unclear to the referee, calcium is indeed included in all liposome-binding assays at [5mM] as described in the methods since we have previously shown that mutation of the calcium ligands or replacement of calcium with magnesium prevents liposome binding for Jagged ligands (as would be predicted from the structural work; Chillakuri et al 2013).

• Giving the clinical relevance of mutations in the C2 domain of Jagged1 and EHBA, if the authors did their cellular signaling assays (Fig 5B) in a hepatic cell line do they think the mutations might have a stronger effect in these reporter assays? Really a comment more than a question.

The most relevant cell type would be a cholangiocyte but unfortunately we do not have access to a suitable line to construct a reporter system.

• In the authors' receptor-ligand plate binding assay, the receptors are pre-clustered via neutravidin. Is pre-clustering required in order to see binding in this assay?

Binding can be detected in the absence of pre-clustering when a high Notch concentration and longer incubation times are used. However, this method is far less reproducible, involves additional incubation and washing steps and is overall much less satisfactory so is not the method of choice.

• It is not clear from Figure 4A what binding to low and high protein concentrations refers to, ligand or receptor. The methods mention 0-300nM Notch is used in the assays. Can the authors clarify how this assay is performed?

We apologise and have clarified the legend and methods. The high (H) and low (L) refer to Notch concentrations of 300 nM and 20 nM which are selected from the concentration range used to evaluate binding.

• Also regarding Figure 4A, there are no negative controls, e.g. a ligand binding mutant or an EGF construct not involved in ligand-receptor complexes, and because of this, it is not clear to me whether binding is observed at the "low" protein concentrations or not.

We have added in data from the same experiments for DLL3, which is structurally highly similar but does not bind Notch, as a negative control.

• Is there a mistake in the asterisks in Figure 4C that denote statistical significance? I would have thought that the pairwise comparisons should be with the first lane (J1+/N1+). Is this correct?

No – we wanted to show significance w.r.t. the ‘boosted’ interaction since the other columns are reduced in that effect, the annotation is therefore as we intended. We agree there are many ways in which a statistical analysis of such complex data could be presented, but we hope that the one we have chosen helps to illustrate the key findings of the experiment.

• I find the binding result for the EHBA mutation R203K rather striking (Figure 5C). Could the authors include a structural basis for this seemingly conservative change?

As we note in the main text and in figure 5A R203 lies directly in the Notch binding site. It is therefore unsurprising to us that substituting it with a lysine that bears only a single NH₂ on its side chain and is also a different length has a profound effect on binding. We have added a very brief note of this in the text:

As predicted, since it involves replacing the larger Arg with a shorter Lys, the disease-causing variant within the Notch-binding site...

2nd Editorial Decision

30 March 2017

Thank you for submitting a revised version of your manuscript. The manuscript has now been seen by one of the original referees, who finds that their main concerns have now been addressed. There are just a few minor issues to be dealt with before formal acceptance here. Congratulations on a nice study!

- 1) Appendix needs a table of content.
- 2) Please submit Table 1 as a doc or xls file.
- 3) Please update references according to The EMBO Journal style (<http://emboj.embopress.org/authorguide#referencesformat>).
- 4) Figure legends for figures 4, 5, S2: please include the number of replicates used and statistic methods applied.
- 5) Our journal has Expanded View section for data that are not essential for the main message of the manuscript. Since your article contains only two supplemental figures, which are currently in the Appendix section, I recommend transforming these figures in Expanded View figures to increase their accessibility. For this purpose, please rename the figures Figure EV1 and Figure EV2, update callouts in the text and upload as separate image files similarly to the main figures (please see our author guidelines: <http://emboj.embopress.org/authorguide#expandedview>).

Papers published in The EMBO Journal include a 'Synopsis' to further enhance discoverability. Synopses are displayed on the html version of the paper and are freely accessible to all readers. The synopsis includes a short introductory paragraph - written by the handling editor - as well as 2-5 one-sentence bullet points that summarise the paper and are provided by the authors. I would therefore ask you to send us your suggestions for bullet points.

Please also provide an image for the synopsis. This image should provide a rapid overview of the

question addressed in the study, but still needs to be kept fairly modest, since the image size cannot exceed 550x400 pixels.

REFEREE REPORT

Referee #2:

The authors have satisfactorily addressed all of my criticisms/concerns and I congratulate them on an interesting manuscript.

2nd Revision - authors' response

06 April 2017

The authors made the requested changes and submitted the final version of their manuscript.

3rd Editorial Decision

07 April 2017

Thank you for providing the final changes to the manuscript. I am now pleased to accept your manuscript for publication in the EMBO Journal.

Corresponding Author Name: SUSAN LEA

Journal Submitted to: EMBO J.

Manuscript Number: EMBOJ-2017-96632R